# Peer review of "Spectral Riemann Surface Topology of Gapped Non-Hermitian Systems"

_SciPost Physics_

## Round 3 · Author Response

Warnings issued while processing user-supplied markup:
- Inconsistency: Markdown and reStructuredText syntaxes are mixed. Markdown will be used.
Add "#coerce:reST" or "#coerce:plain" as the first line of your text to force reStructuredText or no markup.
You may also contact the helpdesk if the formatting is incorrect and you are unable to edit your text.
Response to the Referee 1
We thank the referee for their review of our manuscript.
They have judged our manuscript to address “the highly topical field of topology in non-Hermitian physics” and to propose an interesting and “very nice” parallel between topological features of the toric code and the band structure of non-Hermitian Bloch Hamiltonians. Their critique focuses on the accessibility to “random readers” and the “general theoretical audience” as well as a lack of explicit examples to illustrate the ideas proposed.
Before addressing these points, in which regard we have made significant modifications and expansions to our manuscript, we would like to state the following:
We do not believe that the report reflects a sufficient understanding of the foundations of the topics discussed and leaves questions regarding the understanding of the format for which this manuscript was submitted.
The referee states that the manuscript “operates with specific terms, most of which are not properly defined or explained in the text.”
The only further specification raised regarding this point concerns the notion of energy Riemann surfaces, “already [in] the very first paragraph”. They grant that “of course, the general theoretical audience should know what Riemann surfaces are” — namely connected 1D complex manifolds, ubiquitously tied to multivalued complex-valued(!) functions. But their following critique of Fig. 1, which shows a relevant configuration of such surfaces, asks the question of why real part and absolute value of such surfaces would differ (this difference between real part and absolute value is certainly to be generically expected for any complex-valued function). They further explicitly state that already this “enhances [their] confusion” regarding “notions and objects discussed in the manuscript”. Consequently, they question the clarity of Fig. 4, stating it “appears as an illustration […] showing that the mug with a handle is topological equivalent to a donut” and that it is “unclear why [sic.] Fig.4 has to do with Eq.(5)”. The equation in question states a Bloch Hamiltonian, yet it appears surprising to the referee that an illustration of a system, whose parameter space is the two-dimensional Brillouin zone, should have toroidal structure (“a donut”).
Moreover, the referee critiques that no efforts were made to make the article “self-contained” (!), while pointing out the inclusion of “all those references” provided. We certainly understand, and fully agree, with the aim for accessibility and the importance of intelligibility and reproducibility.
Self-containedness, in the sense of reintroducing base concepts which “the general theoretical audience should know” such as the notion of a Riemann surface or the Brillouin zone, is however neither customary nor purposeful for regular research articles — this manuscript is not a review, it is submitted for consideration as regular research article.
Regardless of this statement, we have taken seriously the points raised to improve the accessibility of our manuscript and have addressed them as follows:
(1) Paragraph length.
Upon suggestion of referee 2, we have restructured the manuscript to first focus on the non-Hermitian model and then explaining the analogy to the toric code. In doing so we have in particular taken care to expand paragraphs into smaller sections.
(2) Figures and explicit examples.
To clarify our discussion, we have included further concrete examples in Sec. 2 (Non-Hermitian Bloch Hamiltonians and spectral Riemann surfaces) and Sec. 3.1 (Closed Fermi cuts). Their relation to Fig.1 and a new Fig.2, showcasing the formation of Fermi cuts, have been addressed in greater detail. In addition, we have included Sec. 3.3 (Time-reversal-symmetric non-Hermitian systems), which provides further clarification on the underlying structure in the central model presented in the manuscript (Eq.(9), formerly Eq.(5)).
A complete list of changes made can be found in the following. A redline version of the updated manuscript is enclosed for convenience.
Response to the Referee 2
We thank the referee for their attentive review of our manuscript. They have raised multiple points, which we would like to address as follows.
(3) “The authors start explaining the toric code, and later moving towards their non-Hermitian case. From my perspective it would be more clear if they make a complete explanation of their non-Hermitian case, and then later on they show the analogy with the toric code”
We have restructured our manuscript following this suggestion and agree that it has helped to provide a more accessible presentation.
(1) “I would encourage the authors to expand some paragraphs in smaller sections, more clearly outlining the important steps in the calculation.”
In rearranging the manuscript, we have taken the opportunity to expand upon the discussion in smaller paragraphs. These include in particular the addition of concrete examples that showcase and discuss features of the introduced Fermi cuts, as well as a section (3.3) clarifying the underlying structure of the central model presented in greater detail.
(2) “If a similar analogy could be done with a 1D non-Hermitian model, I would encourage the authors to include it as it can be greatly helpful to understand their idea. The physics of that case would be of course different from the toric code, yet I believe that it would substantially clarify the essence of their manuscript.”
Such a model unfortunately lacks a central structure, which enables the formation of gapped spectra within the context of our discussion: the presence of crossing points on the Brillouin zone boundary for closed Fermi arcs which do not give rise to EPs. To highlight this property in the two-dimensional model, we have included a new section (3.3) and expanded explanations throughout the text.
(4) “Section 2.3 would benefit from a concrete example, probably accompanied with a figure.”
To provide illustrative examples of the multi-band extension discussed in this section, we have expanded the discussion of the three-band case by an additional figure (Fig.6). It showcases how Fermi cuts between different sheets can be overlaid and how higher-order EPs, which close the complex energy gap, may arise as a result.
- In addition, we have expanded the Implementation section through a more detailed discussion of a realisation within acoustic metasurfaces. This includes a schematic visualisation of the structure of such a surface, as well as comments on how the measurement of the dispersion grants access to the Fermi-cut information.
A complete list of changes made can be found in the following. A redline version of the updated manuscript is enclosed for the referees' convenience.

---

## Round 3 · List of Changes

List of changes made:
(entire manuscript) Rearranged sections such that the study focuses on the introduction of the non-Hermitian models first and later discusses the analogy to the toric code.
(p2&Fig.1) Amended or changed formulations in introductory discussion of Riemann surfaces of the complex-valued energy function in non-Hermitian systems to improve clarity.
(p3) Added concrete example, which highlights EPs, to improve accessibility for a wider audience.
*(p4) Added segue section for Sec. 3 in line with rearrangement of manuscript.
*(p4,5) Added concrete example (including discussion and new Figure 2), which illustrates the formation of closed Fermi cuts.
*(p6) Adjusted Sec. 3.2 to fit rearrangement of the manuscript.
*(p7) Added Sec.3.3, which expands upon the underlying structure for the formation of closed Fermi cuts, in order to clarify the following discussion of the central model presented in our text.
*(p7, sec 4) Added segue section of Sec. 4 in line with rearrangement of manuscript.
*(p9 Fig.5(b), & p6 Fig.3) Split previous figure in line with the rearrangement of the manuscript. Adjusted captions accordingly.
*(p9) Added comment, which clarifies the structure of the model Hamiltonian in reference to the added section 3.3
*(p10 & Fig.6) Added examples for a three-band system to exemplify the discussion of multi-band generalisations and highlight the formation of higher-order EPs, which close the complex energy gap.
*(p11,12&Fig.7) Expanded discussion of Implementations section, including a schematic figure of the relevant acoustic metasurface and comments on how to access information on Fermi cuts from measurements.
*(p12) Added statement to conclusions to reflect the details added in Sec. 3.3

---

## Editorial Decision

unknown